# HIDRA2: deep-learning ensemble sea level and storm tide forecasting in the presence of seiches – the case of Northern Adriatic

Marko Rus[1], Anja Fettich[2], Matej Kristan[1,★], and Matjaž Ličer[2,3,★]

[1]Faculty of Computer and Information Science, Visual Cognitive Systems Lab, University of Ljubljana, Ljubljana, Slovenia
[2]Slovenian Environment Agency, Office for Meteorology, Hidrology and Oceanography, Ljubljana, Slovenia
[3]National Institute of Biology, Marine Biology Station, Piran, Slovenia
★These authors contributed equally to this research and share last authorship.

**Correspondence:** Marko Rus (marko.rus@fri.uni-lj.si)

**Abstract.** We propose a new deep-learning architecture HIDRA2 for sea level and storm tide modeling, which is extremely fast to train and apply, and outperforms both our previous network design HIDRA1 and two state-of-the-art numerical ocean models (a NEMO engine with sea level data assimilation and a SCHISM ocean modeling system), over all sea level bins and all forecast lead times. The architecture of HIDRA2 employs novel atmospheric, tidal and SSH feature encoders, as well as a novel feature fusion and SSH regression block. HIDRA2 was trained on surface wind and pressure fields from a single member of ECMWF atmospheric ensemble and on Koper tide gauge observations. An extensive ablation study was performed to estimate individual importances of input encoders and data streams. Compared to HIDRA1, the overall mean absolute forecast error is reduced by $13\%$, while on storm events it is lower by even a larger margin of $25\%$. Consistent superior performance over HIDRA1 as well as over general circulation models is observed in both tails of the sea level distribution: low tail forecasting is relevant for marine traffic scheduling to ports of northern Adriatic while high tail accuracy helps coastal flood response. Power spectrum analysis indicates that HIDRA2 most accurately represents the energy density peak centered on the ground state sea surface eigenmode (seiche) and comes close second to SCHISM in the energy band of the first excited eigenmode. To assign model errors to specific frequency bands covering diurnal and semi-diurnal tides and two lowest basin seiches, spectral decomposition of sea levels during several historic storms is performed. HIDRA2 accurately predicts amplitudes and temporal phases of the Adriatic basin seiches, which is an important forecasting benefit due to the high sensitivity of the Adriatic storm tide level to the temporal lag between peak tide and peak seiche.

## 1 Introduction

Global mean sea level rise, related to anthropogenic climate change (Arias et al., 2021), is causing a worldwide increase in coastal flooding frequency (Taherkhani et al., 2020) and is leading to a myriad of negative consequences for coastal communities, civil safety and economies (Ferrarin et al., 2020). Shallow semi-enclosed coastal regional basins like Northern Adriatic (North Central Mediterranean Sea) are facing growing threats of coastal inundation and erosion (Ferrarin et al., 2020), seawater intrusions in freshwater reservoirs and are worsening the conditions for marine traffic. Northern Adriatic ports like Venice,

Koper and Trieste, but also other cultural landmark towns like Chioggia or Piran, have been – or will be – forced to take expensive preventive measures to mitigate their exposure.

The problem of sea level forecasting on the Northern Adriatic Shelf (see Fig. 1 for the shelf location and depth) is two-fold: (i) high sea levels lead to severe flooding of densely populated coastal towns, while (ii) low sea levels may effectively inhibit large marine cargo due to very shallow depths (often below 15 meters) of marine waterways on the shelf and especially in the Gulf of Trieste. Reliable forecasting of both tails, high and low, of sea level distribution is therefore imperative for services like civil safety and cargo scheduling activities in local ports.

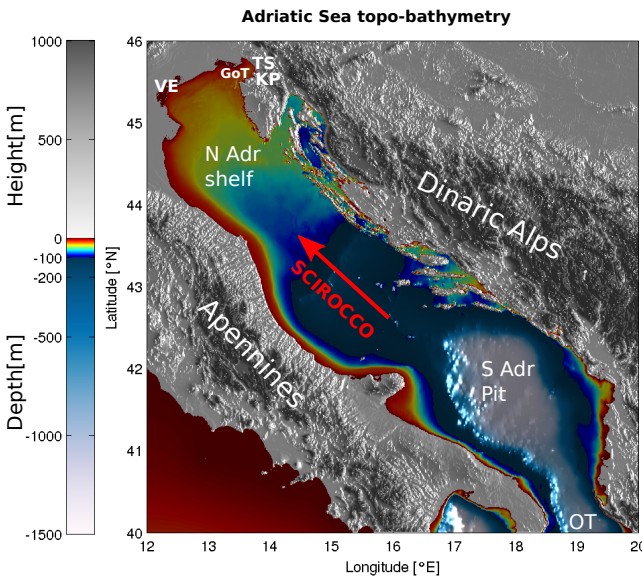

**Figure 1.** Topography and bathymetry of the Adriatic region. Abbreviations used on the map are as follows: TS - Trieste, KP - Koper, GoT - Gulf of Trieste, VE - Venice, N Adr Shelf - Northern Adriatic Shelf, S Adr Pit - Southern Adriatic Pit, OT - Otranto Strait. The direction of Scirocco is marked with the red arrow. The image was created by the authors based on EMODnet bathymetry data, available at https://portal.emodnet-bathymetry.eu/ (last access: 8 June 2022) and Copernicus European Digital Elevation Model, available at https://land.copernicus.eu/imagery-in-situ/eu-dem/eu-dem-v1-0-and-derived-products/eu-dem-v1.0 (last access: 8 June 2022).

The two distribution tails, however, represent two dynamically separate problems. High sea levels always occur due to intense pressure lows and corresponding strong winds during cyclonic activity in the basin, while extremely low sea levels typically occur through a combination of prolonged periods of high atmospheric pressure and spring tides.

Equilibrium ocean response to slow changes in air pressure is captured by the inverse barometer effect, while the wind set-up of the sea level occurs through the vertical momentum flux across the air-sea interface. Dominant winds in the Adriatic basin are

southeasterly Scirocco, blowing along the major axis of the basin (see Fig. 1), and the north-easterly cross-basin Bora. Strong Scirocco events lead to severe storm surges, excitation of basin seiches (Bajo et al., 2019) and potentially severe flooding in the Northern Adriatic. Adriatic along-basin seiches have eigenperiods of 21.5 hours (fundamental eigenmode) and 10.9 hours

(first excited eigenmode), see e.g. (Medvedev et al., 2020), and decay on the timescale of days, mostly due to radiation losses through Otranto (Cerovecki et al., 1997).

In this paper we will adhere to the terminology, proposed in (Gregory et al., 2019): (i) the term *sea level* will denote total time-varying local water depth at the tide gauge in Koper, (ii) the term *sea surface height* is the height of sea level above (or below) the reference ellipsoid, (iii) the term *storm surge* denotes the elevation or depression of the sea surface with respect to the predicted tide during a storm, and (iv) the term *storm tide* is the sea surface height, elevated during a storm by a storm surge.

The key difficulty of sea level forecasting in the Adriatic basin arises from high sensitivity of total sea level to the phase lag between the gravitationally generated tides (independent from meteorological forcing) and meteorologically generated basin seiches (independent from gravitational forcing). This sensitivity can translate reliable atmospheric forecasts with very limited errors in timing and trajectory of the cyclone into substantial errors in the sea level forecast. Probabilistic ensemble forecasting of sea level envelopes with error variance estimation (Žust et al., 2021; Ferrarin et al., 2020; Bernier and Thompson, 2015; Mel

and Lionello, 2014) was therefore explored to tackle this drawback. However, ensemble sea level forecasting is numerically expensive, requires specialized expensive hardware and introduces delays in prediction. To avoid the high numeric cost of ensemble sea level forecasting, computationally efficient machine-learning-based ensemble models have recently been explored (Žust et al., 2021). While these models require a substantial amount of training data to learn the complex relations for reliable predictions, the inference is numerically cheap. For example, single-point Koper sea level predictions from the neural network

HIDRA1 ensemble (Žust et al., 2021) are a million times faster than the full-basin operational NEMO ocean (Madec, 2016) at Slovenian Environment Agency. It is true that HIDRA1 computes prediction for a single variable in a single point, while ocean models compute 4-dimensional evolution of a broad set of oceanic properties but in the operational environment. Faster model prediction times however come with immediate benefits for downstream warning issuing and civil rescue operations.

    Machine learning has thus been explored by several research groups for single-point sea level forecasting. The early ap-

proaches (Imani et al., 2018) were based on classic machine learning models such as support vector machines (Vapnik, 1999) with radial basis function kernels. In their work, Pashova and Popova (2011) and Karimi et al. (2013) utilized shallow fully connected neural networks, but due to simplistic network architectures that did not utilize the numerical atmospheric forecast, they could only report the desired accuracy for short temporal horizons. Ishida et al. (2020) used long short-term memory (LSTM) (Hochreiter and Schmidhuber, 1997) networks together with several atmospheric variables to improve one-hour pre-

diction into the future but did not expand the prediction horizon. Braakmann-Folgmann et al. (2017) predicted further in time by applying a combination of LSTMs and convolutional neural networks, but at a very coarse level. Autoregressive neural networks were considered in (Hieronymus et al., 2019) to increase the temporal resolution and the prediction horizon. Most recently, a convolutional neural network HIDRA1 (Žust et al., 2021) with a specialized architecture to utilize atmospheric data, sea surface heights and astronomic tides was proposed. To the best of our knowledge, HIDRA1 is currently the most accurate

machine-learning sea surface height prediction model with a several days long prediction horizon at an hourly resolution. But while HIDRA1 performed favorably in comparison to the NEMO model used in that study, it failed to beat the NEMO setup at very high and very low ends of sea level distributions. In other words, extreme sea levels (coastal floods on one end and very

low sea levels on the other), which interest us the most, were not yet captured with sufficient reliability and present an open challenge for machine learning methods.

In this paper we propose HIDRA2, our latest attempt at sea level forecasting using deep learning. In contrast to the previous version, HIDRA2 presents a novel architecture with new atmospheric, tidal and SSH feature encoders, as well as a novel feature fusion and SSH regression block. An additional conceptual novelty is that HIDRA2 predicts the full SSH rather than the residual (i.e., the difference between SSH and astronomic tide) as is the case for HIDRA1. The new model extracts relevant information from different spatial locations in the atmosphere signal and predicts the SSH with a three-days horizon at an
unprecedented accuracy, outperforming HIDRA1 as well as two state-of-the-art ocean models.

    The paper is organized as follows. Section 2 introduces sea level and atmospheric model data and performance measures used in our analysis. Section 3 details the new HIDRA2 architecture and the numerical ocean model setup, used as the performance baseline. Section 4 reports the analysis of the HIDRA2 architecture (including an extensive ablation study) and provides detailed quantitative as well as qualitative comparisons with the state-of-the-art numerical ocean models. Conclusions and
outlook are drawn in Sect. 5.

## 2   Training and evaluation datasets

### 2.1   Sea level training data

Sea surface height (SSH) observations during the period 2006–2018 were retrieved from the Koper Mareographic Station ($45° \, 33'$ N, $13° \, 44'$ E; see Fig. 2 for location), which is maintained by the Slovenian Environment Agency (ARSO) and is part
of the European Sea Level Network (Pérez Gómez et al., 2022). Observations are obtained in 10-minute intervals, using both a float-type sensor and an additional radar sea level sensor, and they undergo subsequent quality control at ARSO (Pérez Gómez et al., 2022). Hourly data points are extracted to get the signal used in HIDRA2 training and evaluation.

    The tidal signal in the sea level is independent of atmospheric processes and can be computed by tidal analysis and prediction models. The tidal contribution to Koper SSH considered in this study is estimated from hourly instantaneous SSH values in
one-year segments using the UTIDE Tidal Analysis package for Python (Codiga, 2011). The total sea level series is then decomposed into a tidal and a residual part, where we define the sea level residual as the arithmetic difference between the total sea level and the tidal sea level. According to the ARSO operational protocol, the SSH is classified as a flood if it is higher than $300 \, \mathrm{cm}$. Floods thus constitute $0.41 \, \%$ of all training data.

### 2.2   Atmospheric training data

Atmospheric input used for HIDRA2 training was retrieved from the European Centre for Medium-Range Weather Forecasts (ECMWF) Ensemble Prediction System (Leutbecher and Palmer, 2007). The ECMWF ensemble forecasts come as a global atmospheric ensemble of 50 members with a $0.125°$ arc degree spatial resolution and a 3-hour temporal resolution. The training dataset used in this study consists of (i) 10-meter winds and (ii) mean sea level air pressure from a single fixed (42nd)

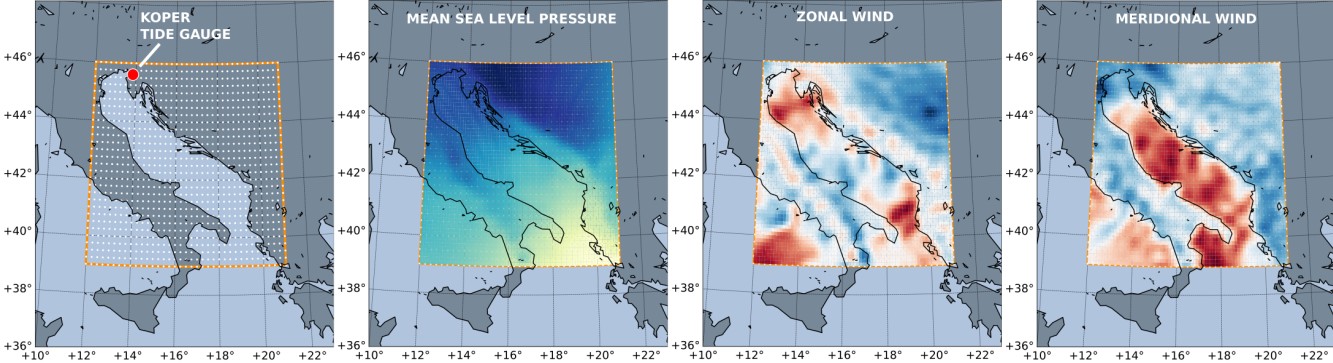

**Figure 2.** HIDRA2 input domain and dataset. The leftmost panel depicts ECMWF grid (white dots) and Koper tide gauge location (red circle). Three panels on the right depict snapshots of ECMWF atmospheric fields used during training.

atmospheric ensemble member during the period 2006–2018. Number 42 was chosen randomly to the extent that it is a tribute to the ultimate answer from the Hitchhiker's Guide to the Galaxy (Adams, 1979). Of course, over multi-year time intervals, this member is completely statistically equivalent to random use of any other member of the ECMWF ensemble prediction system. In other words, we could use any other ensemble member – or choose a different random member each run – without substantially affecting the results. All ECMWF input fields were standardized and cropped to the Adriatic basin, represented by a $57 \times 73$ spatial grid (see Fig. 2). The forecasts were linearly interpolated to hourly timesteps to match the SSH temporal resolution. To simplify the training protocol, a single atmospheric sequence is constructed by concatenating the first 24 hours of daily consecutive ECMWF forecasts into the final atmospheric sequence used in training. HIDRA2 does not require explicit annotation of whether a location point belongs to land or sea, thus land masks are not generated.

## 2.3 Evaluation data

The evaluation input dataset for both HIDRAs and NEMO is disjoint from the training dataset (years 2006–2018) and consists of ECMWF atmospheric predictions and Koper sea levels between 01 June 2019 and 31 December 2020. This period was chosen due to challenging conditions and unusually high incidence of floods. We use the ECMWF daily predictions, each containing 50 ensemble members with three-days prediction lead time. The data are standardized and the dimensionality of the atmospheric data is reduced in the same fashion as described in the Sect. 2.2, except that for inference, the full (i.e. containing all ensemble members) ECMWF three-day forecast is presented to the model. The floods represent $1.1\%$ of the test dataset.

## 2.4 Performance measures

Standard measures, i.e., the mean absolute error (MAE), the root mean square error (RMSE) and the model bias are used to evaluate prediction performance. To reflect the practical suitability, we additionally calculate the prediction accuracy as a ratio between the predictions which are within $10\,\mathrm{cm}$ from the ground truth and all predictions. This $10\,\mathrm{cm}$ threshold reflects an

acceptable deviation from the ground truth and was determined through discussion with the operational forecasting service at ARSO. The metrics are calculated globally by considering all prediction points, as well as separately only on floods to reflect the prediction performance at these critical rare events.

To further probe the flood event prediction capabilities, we make use of the standard performance measures from detection literature: precision $Pr$, recall $Re$ and the F1 measure $F1$. Firstly, we need to define the flood event and then define the notion of the event being detected. Both of these have been defined in discussion with operational forecasters at ARSO. The anchor (i.e., temporal point) of a flood event is defined as the time of the local maximum in an SSH sequence above 300 cm. If the predicted flood event anchor is within a 3 h margin (before or after) from the nearest ground truth flood event anchor, it is considered a true positive $TP$, otherwise it is a false positive $FP$. A flood event in the ground truth is considered a false negative $FN$ if there is no matching flood event anchor in the predicted SSH. Like in the accuracy definition, the tolerance of 10 cm is applied, meaning that predictions below 300 cm are also considered as $TP$ when they appear within the margin of 10 cm, and that false positives with ground truth within 10 cm are ignored. The precision and recall are then calculated as

$$Pr = \frac{TP}{TP + FP} \ , \ Re = \frac{TP}{TP + FN},$$
(1)

while the F1 measure that summarizes the detection performance, i.e.,

$$F1 = 2\frac{Pr \cdot Re}{Pr + Re},$$
(2)

is defined as the harmonic mean between precision and recall.

## 3   Numerical models

### 3.1   HIDRA2

The proposed HIDRA2 is the second generation of a deep neural model for sea surface height prediction, with HIDRA1 (Žust et al., 2021) being the first. The new architecture is shown in Fig. 3. The input data is encoded by three encoders: the wind and pressure sequences for the past 72 h are processed and merged by the Atmospheric encoder (Sect. 3.1.1), the tidal signal for the future 72 h is encoded by the Tidal encoder, and the sea surface height measurements coupled with the tide for the past 24 h are encoded by the SSH encoder (Sect. 3.1.2). The outputs of all three encoders are re-calibrated, fused with the past 72 h SSH and regressed into the final SSH hourly predictions for the future 72 h by the Fusion-regression block (Sect. 3.1.3). A single prediction run of HIDRA2 model creates a 72-hour sea level timeseries for Koper location. Subsections below detail the individual blocks.

### 3.1.1   Atmospheric encoder

The atmospheric data for the Adriatic basin at a given time-step is represented by a $57 \times 73$ spatial grid, i.e., an image. HIDRA2 assumes that coarse spatial resolution of atmospheric data contains enough information to provide satisfactory results, so it first downsamples the atmospheric data to $9 \times 12$ grid by an average pooling operation.

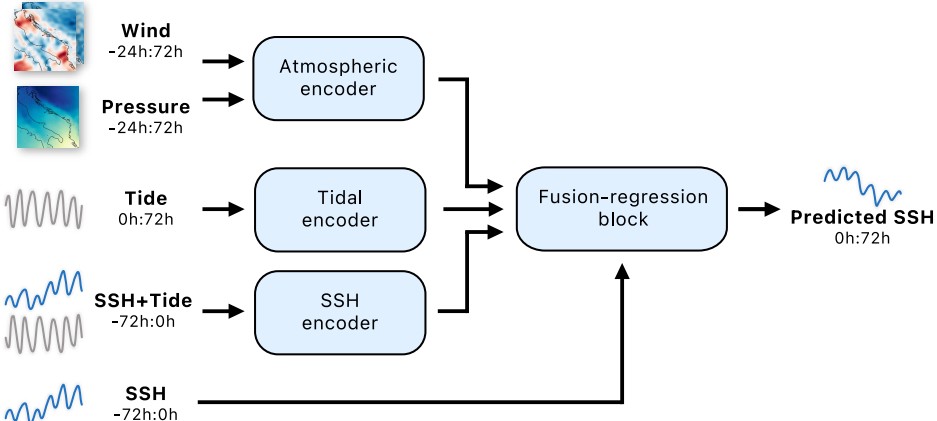

**Figure 3.** The HIDRA2 architecture. The Atmospheric encoder embeds the wind and pressure sequences with learnable temporal subsampling and pattern prototype matching to extract relevant features from different geographic locations and fuse them temporally into a single feature embedding. The Tidal and SSH encoders encode the future tide evolution and the past SSH and tide observations, respectively. All features are re-calibrated, fused with the past 72 h SSH and regressed into the final SSH predictions by the Fusion-regression block. Notation $a$:$b$ indicates hourly data points from the interval $(a, b]$, while the prediction point is at the index 0.

The Atmospheric encoder is composed of two stages. In the first stage (shown in Fig. 4), the sequences of the wind and pressure images are independently processed by their respective encoding blocks, which use the same architecture. The wind image sequence of 96 h (the past 24 h and future 72 h) is divided into 24 groups of four consecutive hours, which are processed independently. The spatial and temporal dimension of each group is reduced by a learnable 2D convolutional layer with a $3 \times 3$ kernel, stride 2 and 64 output channels[1]. A ReLU activation and Dropout layers are applied, followed by a convolutional layer with 512 $4 \times 5$ kernels, which are by size equal to the input, meaning that convolution is essentially a dot product between each kernel and the input. The operation yields a higher value if kernel is similar to the input, so we refer to it as a *prototype matching layer*. It extracts features from different spatial positions, thus producing a 512-dimensional feature vector per group, i.e., 24 temporal vectors of size 512. The same processing architecture is applied to the pressure image sequence to produce 24 vectors of size 512. The two outputs are then concatenated to form a mixed set of 24 wind-pressure features of size 1024.

The second stage of the Atmospheric encoder (Fig. 5) extracts the temporal atmospheric features by considering the consecutive wind-pressure features extracted by the first stage. A 1D convolutional layer with the kernel temporal dimension size 5 and with 256 output channels[2] is applied, entangling the information from temporal segments equivalent to 20 h in length. Note that because we are using a convolutional layer instead of the fully connected layer, the number of learnable parameters of the entire Atmospheric encoder is independent of the forecast horizon. Each of the obtained 20 features[3] is then independently

---

[1]Note that the number of output channels is equal to the number of different kernels used in the layer.

[2]Note that the sizes of the kernels in the 1D convolutional layer are $1024 \times 5$, but with the first dimension matching the size of the input features, the convolution displacements are only along the second dimension, hence the 1D convolutional direction implied by the layer's name.

[3]The fact that the number of output segments is equal to the 20 h timespan is coincidental.

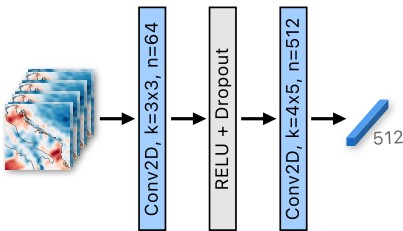

**Figure 4.** The first stage of the Atmospheric encoder. The input are four consecutive hours with two wind channels (this case), or pressure. $3 \times 3$ convolution is applied, followed by a prototype matching layer, outputting a single vector of size 512. Note that 24 independent passes are performed in parallel for the entire atmospheric input sequence. The variables $k$ and $n$ denote the kernel size and the number of output channels, respectively.

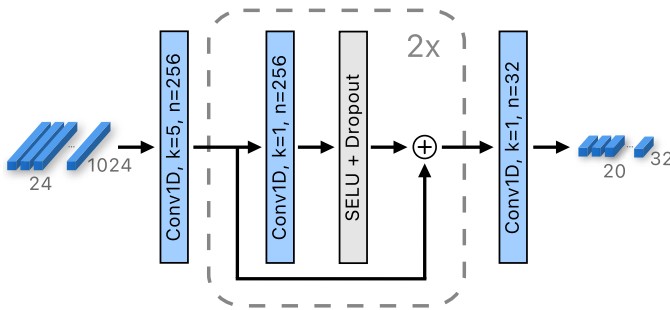

**Figure 5.** The second stage of the Atmospheric encoder. Features from all time points and both wind and pressure are processed with a 1D convolution, followed by two blocks with residual connections. The last convolution reduces feature dimensionality. The variables $k$ and $n$ denote the kernel size and the number of output channels, respectively.

processed by a network containing two blocks of residual connections, each involving 1D convolution with kernel temporal dimension size 1 (i.e., $1 \times 256$ kernels), a SELU activation (Klambauer et al., 2017) and a dropout layer. Finally, each of the obtained twenty 256-dimensional features are convolved by 32 $1 \times 256$ kernels to reduce their dimensionality to $20 \times 32$.

### 3.1.2 Tidal and SSH encoders

Both the tidal and SSH encoders use the same architecture, the only difference is in the size of the encoders' input. Figure 6 depicts the SSH encoder, which takes as the input the past 72 h SSH measurements and the tide concatenated into a $72 \times 2$

input tensor and processes them in a similar fashion to the second stage of the Atmospheric encoder: the input is processed by convolution with 256 $3 \times 2$ kernels, which is followed by two consecutive residual blocks, a subsampling max-pooling layer and a final convolutional layer to reduce the dimensionality of features to $17 \times 16$. The Tidal encoder follows the same architecture, the only difference is that the input is the tide forecast for the next 72 h.

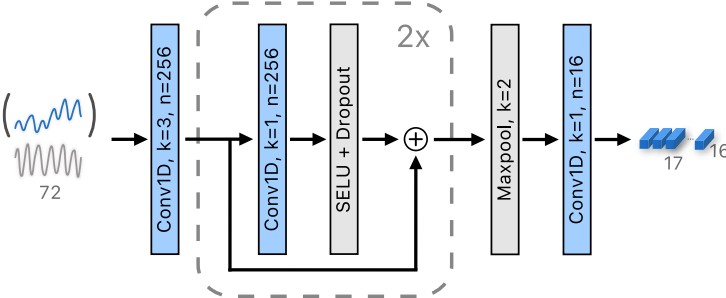

**Figure 6.** The SSH encoder encodes a concatenation of the past SSH and tide by a 1D convolution, followed by two blocks with residual connections, max-pooling temporal reduction and convolution-based feature reduction. The variables $k$ and $n$ denote the kernel size and the number of output channels, respectively.

### 3.1.3 Fusion-regression block

Atmospheric, Tide and SSH encoders produce temporal features of different importances and sizes. To account for that, the features are re-calibrated by normalization with means and variances of the features calculated during training, and then denormalized with learned weights and biases. The form of normalization follows the batch normalization layer (Ioffe and Szegedy, 2015), which applies a 0.9 momentum for updating the running means and variances during training. The normalized features are then concatenated and mixed by a fully connected layer, reducing their final dimension from 1184 to 512 (left part of the

Fig. 7). The obtained 512-dimensional domain context feature vector thus contains rich atmospheric and sea-surface height information from all time points and all parts of the input domain.

While the encoding and mixing operations extract the domain context, the explicit surface height information might not well be retained in the extracted feature vector. To re-inject this information, the obtained domain context feature vector is concatenated with the timeseries of past observed SSH before passing to the final regression block. The latter is composed of

190 two fully connected layers with 584 units, SELU activations and residual connections, followed by a fully connected layer with 72 outputs for the 72 h prediction horizon (see Fig. 7).

### 3.1.4 The network training

HIDRA2 is trained end-to-end using mean squared error (MSE) loss between the predictions and the ground truth. We train the model using the AdamW optimizer (Loshchilov and Hutter, 2017) with standard parameter values (learning rate 1e−4, and

195 the running average damping parameters $\beta_1 = 0.9$ and $\beta_2 = 0.999$), and apply the cosine annealing (Loshchilov and Hutter, 2016) learning schedule to gradually reduce the learning rate during training to 1e−5. The training batch size is set to 512 data samples, and the model is trained for 40 epochs. Training takes approximately 1.5 hours on a single computer with NVIDIA Geforce RTX 2080 TI graphics card, while the inference of a single 72 h prediction for one member of the atmospheric ensemble takes only 4 ms.

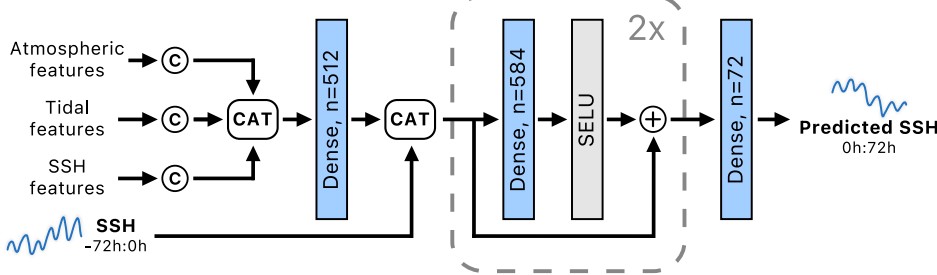

**Figure 7.** The Fusion-regression block firstly re-calibrates the features (the *C* symbol), then concatenated features are passed to a dense layer, which fuses features and reduces their dimensionality. Undistorted SSH is appended and processed with two residual blocks. The final dense layer outputs the predictions. The variable $n$ denotes the number of output channels.

### 3.1.5 Summary of differences to HIDRA1

While there are many differences between HIDRA2 and HIDRA1, we summarize only the major conceptual ones for a clearer exposition of the contributions. HIDRA1 uses wind, pressure and $2$ m temperature from ECMWF predictions, while our preliminary study showed that the new HIDRA2 architecture does not benefit from the temperature, thus only wind and pressure are considered. HIDRA1 concatenates all atmospheric inputs at a timestep and encodes them by Resnet (He et al., 2016) blocks. While Resnet excels in computer vision tasks that rely on high-level semantic feature abstraction, we argue that tailored shallower encoders are more appropriate for the extraction of meaningful atmospheric patterns. HIDRA2 thus separately encodes the wind and pressure by shallow encoders, which apply spatial pattern features extraction, and then mixes the features from the two atmospheric variables by extracting temporal patterns. While this allows HIDRA2 to extract multiple spatial patterns in the data, only a single set of spatial weights is used to fuse the atmospheric features at a given time-step in HIDRA1, consequently reducing its expressive power. HIDRA1 first averages four-hour atmospheric input data to temporally subsample the input, while HIDRA2 considers per-hour inputs and learns the appropriate spatio-temporal subsampling to maximize its predictive power. Another advantage of HIDRA2 is that it encodes the SSH input and mixes it with the atmospheric features early in the network to create a domain context feature vector before the final regression, while HIDRA1 considers only the atmospheric data for the context vector. Finally, HIDRA1 predicts the SSH residual (i.e., the difference between SSH and the astronomic tide), while HIDRA2 directly predicts the full SSH.

### 3.2 Ocean models

In this section we briefly describe two different numerical ocean modeling setups used for benchmarking HIDRA2. The two setups differ in several important respects. One is based on NEMO ocean engine (Madec, 2016) and the other on SCHISM (Zhang et al., 2016) modeling environment. NEMO setup is described in more detail in Sect. 3.2.1 and it is a forecasting setup. SCHISM setup is described in Sect. 3.2.2 and it is a reanalysis setup (Toomey et al., 2022). For brevity we will refer to the two setups presented below simply as NEMO or SCHISM.

### 3.2.1 NEMO ocean model

The Copernicus Marine Environment Monitoring Service (CMEMS) product MEDSEA_ANALYSISFORECAST_PHY_006-_013 (see Clementi et al. (2021)), was used as one of two numerical baselines for HIDRA2. This product is based on a Mediterranean basin configuration of the NEMO ocean model (Madec, 2016), and provides daily ocean forecasts for sea surface height above the geoid, temperature, salinity, circulation and mixed layer depth. The model domain spans the entire Mediterranean basin with a $(1/24)°$ resolution and has 141 unevenly spaced vertical levels. The model solutions are operationally constrained to near-real-time observations using a 3D variational assimilation scheme of temperature, salinity and along-track satellite sea level anomaly observations. The atmospheric forcing to the CMEMS model is provided by the ECMWF. Further details about the modeling setup can be found in (Clementi et al., 2021). In this study, an SSH timeseries at the closest point to Koper tide gauge was extracted from the Mediterranean ocean model forecast.

### 3.2.2 SCHISM ocean model

A barotropic setup of SCHISM storm surge and wind-wave modeling environment (Toomey et al., 2022) was used as a second numerical baseline for HIDRA2. In this study, a single SSH timeseries from SCHISM reanalysis (Toomey et al., 2022) at the closest point to Koper tide gauge was extracted and used for comparisons to HIDRA models. SCHISM runs on an unstructured mesh covering the entire Mediterranean basin and extending into the Atlantic ocean in the west. Its lateral boundary is forced by an equilibrium inverted barometer ocean response to atmospheric pressure, while its surface forcing consists of ERA5 surface fields. SCHISM unstructured grid allows for very high coastal resolutions, reaching some 200 m close to the coast.

### 3.2.3 Ocean model offset adjustment

Both NEMO and SCHISM sea levels, denoted here jointly as $y^{\mathrm{model}}$, at any given location reflect departures from the local geoid and hence do not represent the absolute local depth of the water. The latter is furthermore also driven by low-frequency processes on the scales of many weeks or months which are often difficult to capture for regional basin models on synoptic timescales. Prior to benchmarking, model results therefore have to be offset-adjusted to obtain total sea levels (required by port authorities and civil rescue) as follows. A time-averaged model (NEMO or SCHISM) SSH offset $\epsilon_n$ on the $n$-th hour of the forecast day is defined as

$$\epsilon_n = n^{-1} \sum_{k=1}^{n} \left[ y^{\mathrm{model}}(t_k) - y^{\mathrm{kp}}(t_k) \right], \tag{3}$$

where $y^{\mathrm{kp}}(t)$ is the observed Koper sea level.

Each day, the value of $\epsilon_{12}$ is subtracted from $y^{\mathrm{model}}(t)$ to ensure a zero bias for the first 12 h of the model day. Note that, despite this adjustment, the complete 72-hour modeled time-series may still exhibit a non-zero bias. Similar offset adjustment is not required for HIDRAs, since they predict the full SSH and learn to appropriately adjust for the offset automatically.

## 4 Results and discussion

### 4.1 HIDRA2 architecture analysis

As noted in Sect. 2, HIDRA2 was trained on the period 2006–2018 and evaluated on the period between 01 June 2019 and 31 December 2020. For evaluation, a single prediction is obtained by averaging predictions over all 50 ECMWF ensemble members. In the following we analyze the architectural choices of HIDRA2. The prediction of full SSH is justified in Sect. 4.1.1, while Sect. 4.1.2 reports an ablation study that aims to determine the role of specific encoders and types of input data. The results of all experiments are collected in Table 1.

#### 4.1.1 Predicting the full SSH vs the residual

A valid hypothesis can be made that predicting the residual (i.e., the difference between the full SSH and the tide) might be more beneficial than predicting the full SSH, since the network parameters might be better utilized by focusing only on the part of SSH not affected by the astronomic tide. In fact, HIDRA1 (Žust et al., 2021) does exactly this – it accepts and forecasts the residual. To explore this hypothesis, HIDRA2 was modified to predict only the residual (HIDRA2$_{res}$), by replacing the SSH input in the SSH encoder with the residual, as well as in the Fusion-regression block. Results in Table 1 indicate a similar overall performance when only the residuals are considered in HIDRA2. However, considering only stormy periods, we observe a substantial increase of the prediction error ($+17.4\,\%$ MAE). This means that full SSH prediction is very beneficial for predicting floods, while incurring only a small drop in the overall performance.

A possible explanation of this somewhat surprising behavior could perhaps be related to nonlinear interactions between tides and storm surges: both tides and storm surges modify local water depth which impacts their own barotropic wave propagation speeds and topographic amplifications, which ultimately define the onset time and the amplitude of any coastal flood in Koper. Such interactions are non-existent during calm conditions but they do play a role during stormy periods (Ferrarin et al., 2022). Perhaps HIDRA2 is able to anticipate certain aspects of nonlinear tide-surge couplings. This explanation is also consistent with the fact, detailed in Section 4.1.2, that among all atmospherically driven models the de-tided version HIDRA2$_{res}$ shows the worst performance during storm tide events, while versions incorporating tides come closest to HIDRA2 (see Fig. 8).

#### 4.1.2 Ablation study: the importance of encoders and input data

An ablation study was executed to evaluate the importance of individual encoders and input data types. To estimate encoder importances we removed each of the encoders in a separate experiment (and withheld all of their input data, see Fig. 3) and retrained thus obtained ablated network. Ablation training and evaluation were conducted on identical datasets as with HIDRA2: years 2006–2018 represented the training set and the time window between 01 June 2019 and 31 December 2020 served as an independent validation set. Note that regardless of the encoder input, HIDRA2 always receives unencoded SSH data directly into its fusion-regression block (bottom dataflow branch in Fig. 3).

The following encoder ablations were performed:

**Table 1.** Performance of ablated HIDRA2 designs evaluated over all sea level bins (the *Overall* columns) and only on storm tide events (*Storm tide events* column). The evaluation period spans 01 June 2019–31 December 2020, which is completely independent from the training data. The top row shows performance of HIDRA2 predicting the residual, next three rows show performances of encoder ablations, the following three rows correspond to SSH input ablations and re-calibration. The bottom row corresponds to the final version of HIDRA2.

| | Overall | | | | Storm tide events | | | | | | |
|---|---|---|---|---|---|---|---|---|---|---|---|
| | MAE | RMSE | Bias | Acc | MAE | RMSE | Bias | Acc | Re | Pr | F1 |
| modification | [cm] | [cm] | [cm] | [%] | [cm] | [cm] | [cm] | [%] | [%] | [%] | [%] |
| HIDRA2$_{res}$ | **4.11** | 5.84 | -0.69 | 92.77 | 11.47 | 15.52 | -9.13 | 57.20 | 76.16 | 91.27 | 83.03 |
| HIDRA2$_{\backslash atmE}$ | 7.54 | 11.40 | -0.53 | 75.75 | 27.86 | 34.28 | -26.21 | 23.01 | 41.72 | 91.30 | 57.27 |
| HIDRA2$_{\backslash tidE}$ | 4.60 | 6.37 | **-0.02** | 90.66 | 10.57 | 14.82 | -7.74 | 61.94 | 78.81 | 92.25 | 85.00 |
| HIDRA2$_{\backslash sshE}$ | 4.24 | 5.97 | -0.37 | 92.45 | 10.80 | 15.01 | -8.09 | 61.08 | 80.13 | **93.80** | 86.43 |
| HIDRA2$_{\backslash tidI}$ | 4.24 | 5.96 | -0.47 | 92.28 | 10.65 | 14.74 | -7.63 | 58.92 | 80.13 | 93.08 | 86.12 |
| HIDRA2$_{\backslash sshI}$ | 4.23 | 5.93 | -0.36 | 92.59 | 11.04 | 15.14 | -7.99 | 58.49 | 82.78 | 91.24 | 86.81 |
| HIDRA2$_{\backslash norm}$ | 4.14 | 5.85 | -0.03 | 92.72 | 10.38 | 14.35 | -7.48 | 60.43 | 81.46 | 92.48 | 86.62 |
| HIDRA2 | 4.12 | **5.82** | 0.21 | **92.89** | **9.77** | **14.07** | **-5.99** | **64.52** | **84.11** | 91.37 | **87.59** |

1. Removal of the Atmospheric encoder (HIDRA2$_{\backslash atmE}$). Network HIDRA2$_{\backslash atmE}$ obtained no atmospheric input data but it did receive SSH and tidal data.

2. Removal of the Tidal encoder (HIDRA2$_{\backslash tidE}$). Network HIDRA2$_{\backslash tidE}$ obtained no tidal input data for tidal encoding but it did receive SSH and tidal data through the SSH encoder and atmospheric data through the Atmospheric encoder.

3. Removal of the SSH encoder (HIDRA2$_{\backslash sshE}$). Network HIDRA2$_{\backslash sshE}$ received atmospheric and tidal data through Atmospheric and Tidal encoders, but it did not receive any SSH input via the SSH encoder.

The results in the Table 1 show that MAE increases with each modification, particularly during storm events. Removal of the Atmospheric encoder results in the most significant performance drop, indicating that the atmospheric features convey by far the most relevant predictive information. A significant performance drop is observed as well when removing the Tidal encoder. The SSH encoder has the smallest impact on overall performance, yet still importantly contributes to the prediction accuracy during storms.

Two further ablations were then performed regarding the data types of the sea level input data (the SSH and the tide, see Fig. 3) which are considered in the SSH encoder. We retained HIDRA2 with all three of its encoders but provided the SSH encoder with limited sea level input:

1. Removal of the tidal input to the SSH encoder (HIDRA2$_{\backslash tidI}$). In this case the SSH encoder received as input only total sea level.

2. Removal of the SSH input (HIDRA2$_{\backslash sshI}$). In this case the SSH encoder received as input only tidal sea level.

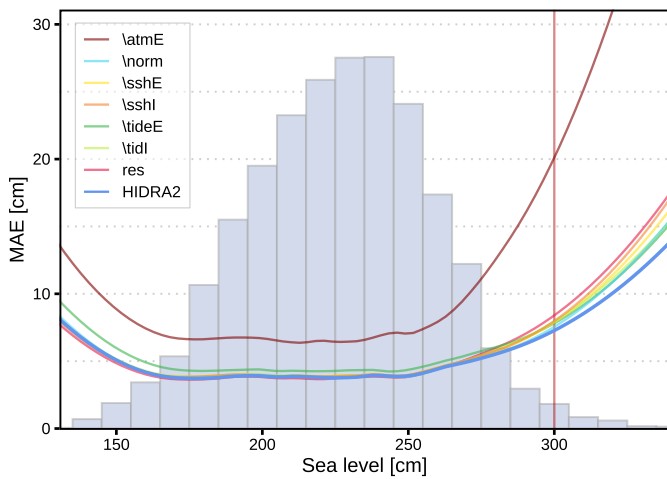

**Figure 8.** Mean Absolute Error (MAE) of ablated HIDRA2 designs evaluated over all sea level bins. Vertical red line indicates the flooding threshold in Piran. Performance of all models was evaluated on a 01 June 2019–31 December 2020 dataset, which is completely independent from the training data.

The results in Table 1 show that the removal of each leads to a consistent but moderate increase of the errors overall. However, the errors increase substantially during storms, indicating the importance of using both types of inputs.

We observe a similar situation when removing the atmospheric and SSH/tide feature re-calibration in the Fusion-regression block (HIDRA2$_{\backslash\text{norm}}$). Results in Table 1 indicate that feature normalization does not affect performance in normal conditions, but it substantially contributes to the prediction accuracy of storm tides. A closer inspection of HIDRA2$_{\backslash\text{norm}}$ showed that the scale of the tidal features is four times larger than the scale of the atmospheric features. Inclusion of the re-calibration blocks, however, remedies this by making the scales of all features (atmospheric, SSH and tidal) approximately the same.

Figure 8 depicts performances of ablated HIDRA2 versions across all sea level bins. Even though most global performance metrics of HIDRA2 (depicted in Table 1) are the best, Fig. 8 indicates that for low sea levels, HIDRA2$_{\text{res}}$ exhibits slightly lower errors. HIDRA2, however, performs substantially better in the flooding regime above 300 cm. This further substantiates our final choice of the HIDRA2 architecture.

### 4.2 Comparison with the state-of-the-art numerical ocean models

HIDRA2 is compared with HIDRA1 (Žust et al., 2021), which is currently the state-of-the-art in machine-learning SSH prediction (Sonnewald et al., 2021), and with state-of-the-art numerical ocean modeling setups NEMO (Madec, 2016) and SCHISM (Toomey et al., 2022). The methods are evaluated on an independent time window (01 June 2019–30 December 2020) and with respect to different SSH values (see Sect. 4.2.1), Sect. 4.2.2 reports performance with respect to the lead times, while spectral analysis is reported in Sect. 4.2.3. The last two sections discuss performance on historical storm surge events (Sect. 4.2.4) and the forecast spectral decomposition of these events (Sect. 4.2.5).

**Table 2.** Performance of HIDRA1, HIDRA2, NEMO and SCHISM over all sea level bins (the *Overall* columns) and only during storm tide events (*Storm tide events* columns). Tidal forecast is included for reference. Evaluation period spans 01 June 2019–31 December 2020, which is completely independent from the training data.

| | Overall | | | | Storm tide events | | | | | | |
|---|---|---|---|---|---|---|---|---|---|---|---|
| | MAE | RMSE | Bias | Acc | MAE | RMSE | Bias | Acc | Re | Pr | F1 |
| | [cm] | [cm] | [cm] | [%] | [cm] | [cm] | [cm] | [%] | [%] | [%] | [%] |
| Tide | 13.82 | 18.86 | -5.13 | 47.45 | 55.75 | 59.45 | -55.75 | 0.00 | 0.00 | / | / |
| NEMO | 6.54 | 8.52 | -1.23 | 79.14 | 13.03 | 17.09 | -11.24 | 49.68 | 63.58 | **100.00** | 77.73 |
| SCHISM | 5.57 | 7.50 | **0.20** | 85.06 | 11.04 | 14.70 | -6.19 | 57.63 | 78.81 | 89.47 | 83.80 |
| HIDRA1 | 4.72 | 6.73 | -0.26 | 90.04 | 12.95 | 17.65 | -10.66 | 53.76 | 74.17 | 94.12 | 82.96 |
| HIDRA2 | **4.12** | **5.82** | 0.21 | **92.89** | **9.77** | **14.07** | **-5.99** | **64.52** | **84.11** | 91.37 | **87.59** |

### 4.2.1 SSH forecast performance

The overall prediction performance and performance restricted to storm events are shown in Table 2. HIDRA2 outperforms HIDRA1, NEMO as well as SCHISM overall as well as during storms, yielding a lower MAE/RMSE and higher accuracy. While HIDRA1 achieves a lower bias, its RMSE/MAE are substantially higher – HIDRA2 outperforms HIDRA1 in MAE by 12.7 % overall, and by 24.6 % during the storm tide events. NEMO achieves the highest precision of flood detection ($Pr = 100$ %), meaning that all detected floods are true positives. But while all NEMO's predicted floods were true, not all floods were predicted, resulting in its low recall of $Re = 63.58$ %. A similar situation is observed for HIDRA1. The recalls for these two methods (NEMO: 63.58 % and HIDRA1: 74.17 %) are substantially lower than that of HIDRA2 ($Re = 84.11$ %), which detects many more floods with fewer false negatives. The excellent trade-off between the precision and recall of HIDRA2 is reflected in its F1 score (87.59 %), which is substantially higher than that of NEMO (77.73 %), HIDRA1 (82.96 %) or the next best SCHISM (83.80 %).

For detailed analysis, we visualize the MAE values of the tested methods with respect to the sea level heights in Fig. 9. HIDRA2 consistently shows the lowest errors at all sea level bins. During storm tides, NEMO outperforms HIDRA1, while HIDRA2 and SCHISM outperform both HIDRA1 and NEMO by several cm. Solid HIDRA2 performance in the low end of the sea level distribution is particularly important to note because of its potentially high significance to marine traffic scheduling in the very shallow seas surrounding the Port of Koper, which is currently restricted to periods of high tides. In summary, HIDRA2 outperforms all state-of-the-art methods for all sea level heights, thus displaying a solid prediction skill in moderate as well as extreme values of the sea surface height.

### 4.2.2 Performance with regard to forecast lead time

We next analyzed how the prediction lead time affects the prediction errors. Figure 10 shows the MAE scores with respect to the prediction lead time for the values between 1 h and 72 h. The MAE of the prediction gradually increases with the lead time

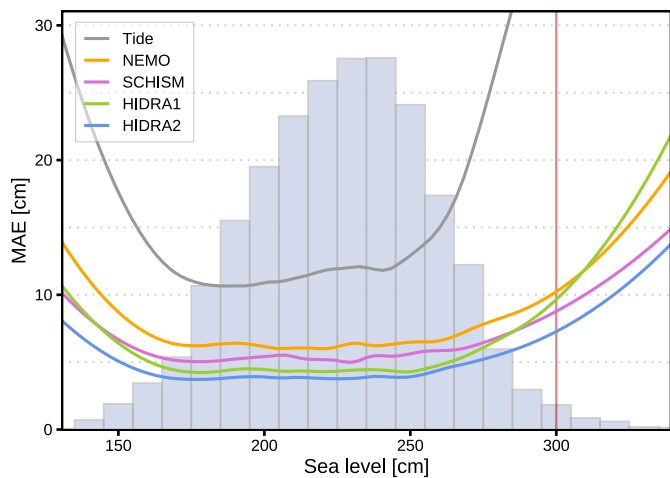

**Figure 9.** HIDRA, NEMO and SCHISM performances with regard to sea level bins (grey histogram in the bottom layer). The coastal flood threshold is marked with a vertical red line. Performance of all models was evaluated on a 01 June 2019–31 December 2020 dataset, which is completely independent from the training data.

for all the tested methods. While overall being a solid performer with MAE well below 10 cm, NEMO exhibits the highest MAE and also the highest MAE variance. Clear signals are observed with 12 h and 24 h periods in the NEMO MAE. Since NEMO includes tides, we suspect this periodicity stems mostly from the errors in either amplitude or phase of the tidal part of the NEMO sea level signal but further research would be necessary to properly substantiate this claim. SCHISM shows better performance (lower MAE) than NEMO but exhibits similar periodicity in errors. Interestingly, while HIDRA2 consistently outperforms HIDRA1 for all lead times, the shapes of the MAE curves show resemblance. While the 12 h period does not seem to be present in the MAE curves of these two models, their 24 h period is clearly present. Further research would, however, be required to substantiate and explain the observed MAE curve behavior.

### 4.2.3 Spectral analysis

To investigate the spectral properties of the modeled and observed SSH timeseries, we computed spectral densities of the HIDRA2, HIDRA1, NEMO and SCHISM predictions. Unless otherwise stated, all time-series analyzed in this section were obtained by concatenating (in time) the first 24 hours of each daily HIDRA2, HIDRA1 and NEMO three-day forecast. Spectral densities (shown in Fig. 11) were then computed as absolute values of a 1D Fast Fourier Transform of the respective series over a fixed frequency domain of $(1 \text{ h})^{-1} - (72 \text{ h})^{-1}$.

Figure 11 indicates that all methods adequately represent the tidal dynamics in Koper. The energy content around the two lowest basin eigenmodes is, however, more discriminatory: NEMO (Fig. 11, left panel) clearly underestimates the spectral density both around the ground state seiche (at 21.5 h period) and around the first excited state (10.9 h period). Similar behavior was noticed in our previous work with an independent configuration of NEMO (Žust et al., 2021). SCHISM, on the

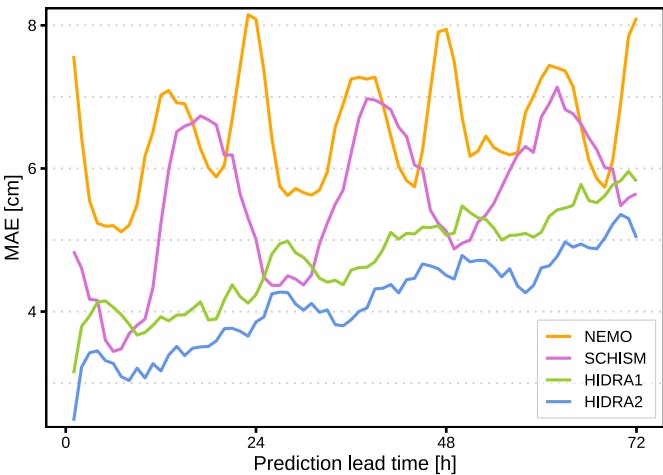

**Figure 10.** MAE score of HIDRA, NEMO and SCHISM models with regard to prediction lead time (between 1 h and 72 h). Performance of all models was evaluated on a 01 June 2019–31 December 2020 dataset, which is completely independent from the training data.

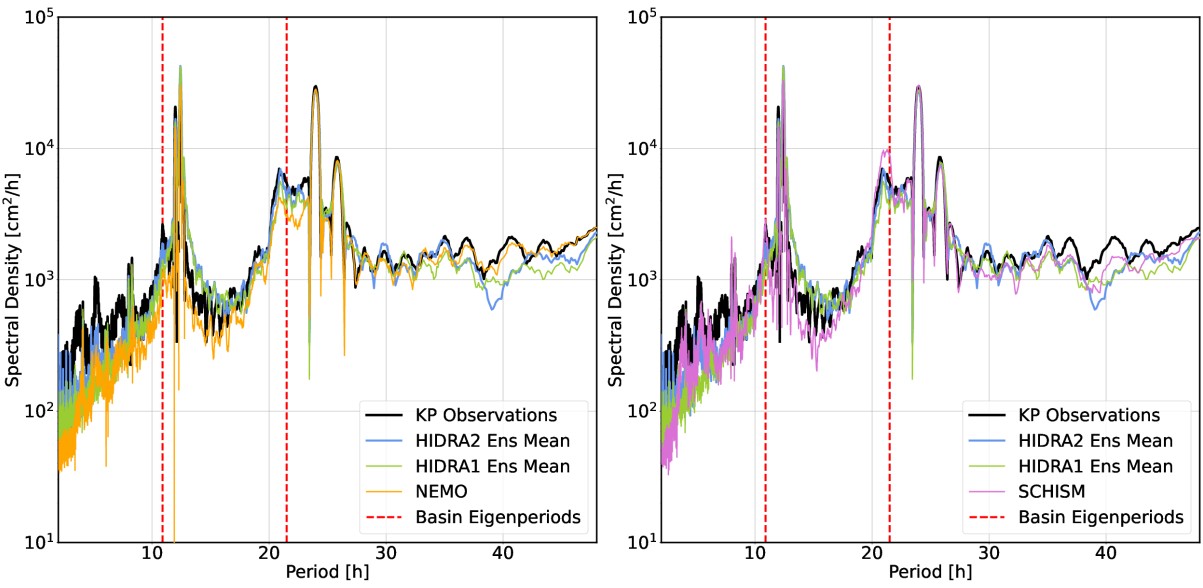

**Figure 11.** Spectral density of SSH timeseries from the Koper tide gauge, HIDRA2, HIDRA1 compared with NEMO (left panel) and SCHISM (right panel), during independent cross-validation time window between 01 June 2019 and 31 December 2020. Sharp peaks at (roughly) 12 and 24 hours indicate the presence of tides, while the two dashed vertical red lines mark the periods of the two lowest Adriatic sea level eigenmodes. For clarity, all plotted spectral densities were filtered using a 3rd-order Savitzky-Golay 24-point window filter.

other hand, overestimates the energy in the ground state seiche band, but reproduces the first excited state energy very well (Fig. 11, right panel). HIDRA1 underestimates the energy of this part of the signal as well, but nevertheless does a bit better

by packing more energy density in these two bands. Predictions of HIDRA2 are clearly the closest to the observations in the ground state seiche band, but come close second to SCHISM around the 10.9 h period.

It appears that HIDRA2 is capable of generating a seiche-like behavior in its predictions. Spectral density, however, discards the temporal component of the signal, and adequate spectral density in the $(21.5 \text{ h})^{-1}$ and $(10.9 \text{ h})^{-1}$ frequency bands says little about whether Adriatic seiches are generated by HIDRA2 at the appropriate times, namely during the storms. To inspect this aspect of HIDRA2 behavior, we now proceed to analyze the predictions during several historic storm tides.

### 4.2.4 Performance during historic storms

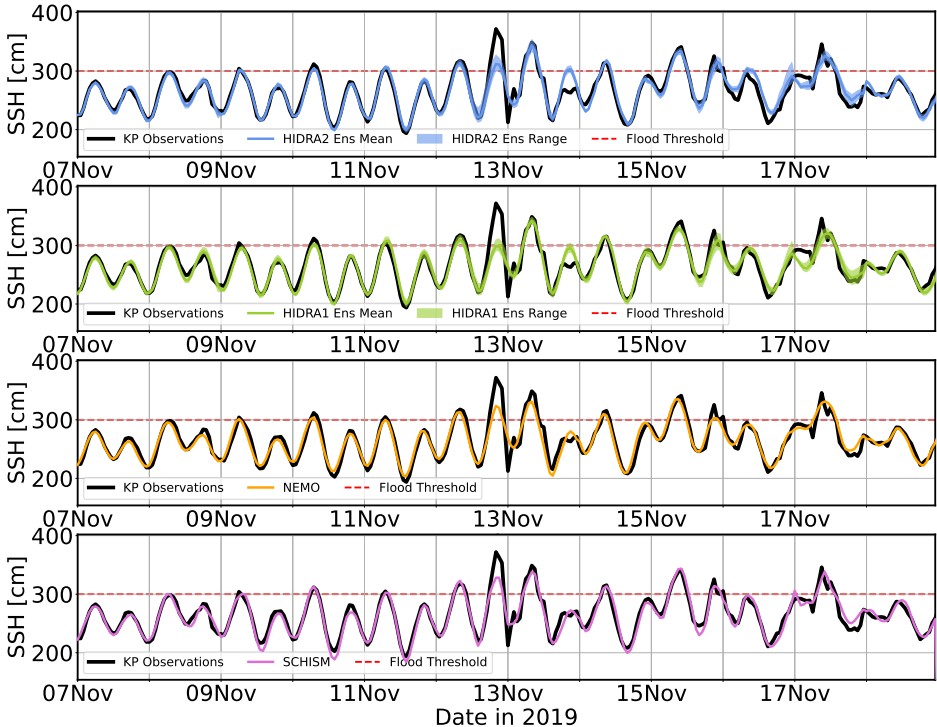

**Figure 12.** Comparison of HIDRA2 ensemble (top row), HIDRA1 ensemble (second row), NEMO forecast run (third row) and SCHISM reanalysis run (fourth row) during the November 2019 flooding sequence in the Northern Adriatic. Semi-transparent regions in the top two plots depict the minimum-maximum envelope of each HIDRA ensemble.

Historic Adriatic storm tide events are used to qualitatively compare the HIDRA2 performance with the state-of-the-art. Storm tides in question occurred during November and December 2019 and were of historic proportions by any criterion. The Slovenian coast was flooded over ten times in a single month and sea levels in Venice were among the highest ever observed. Furthermore, the events in November 2019 turned out to be difficult to model due to the formation of a transient and very localized low pressure over the Gulf of Venice, which went unresolved in most models (Cavaleri et al., 2020). These events,

along with those from December 2019, therefore represent a highly challenging benchmark for any atmospheric model and even more for any downstream SSH prediction method.

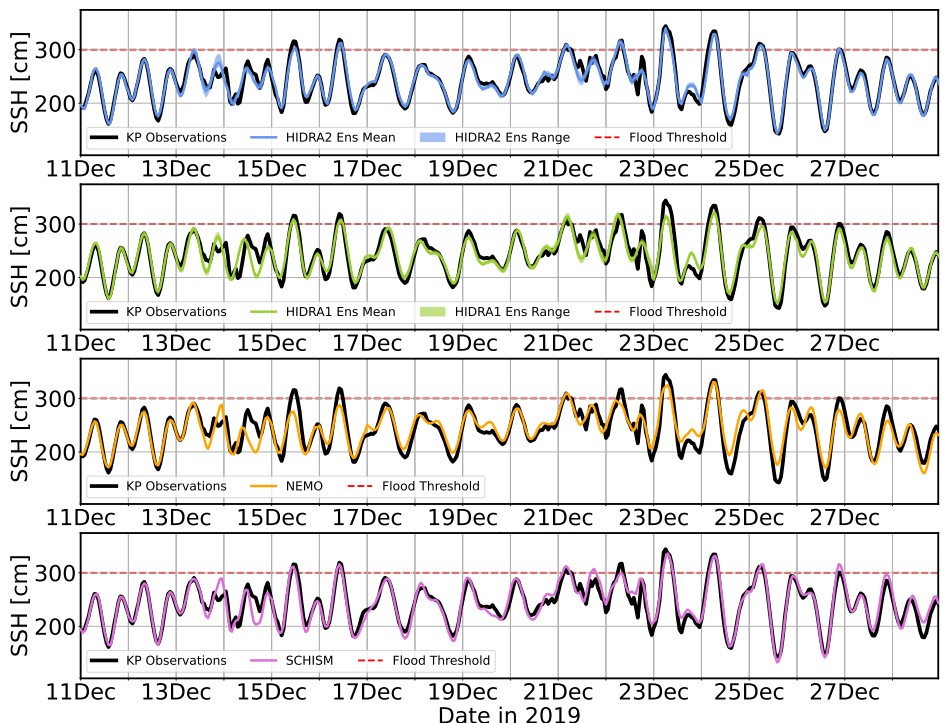

**Figure 13.** Same as Fig. 12 but for December 2019.

Figure 12 shows HIDRA2, HIDRA1, NEMO and SCHISM SSH forecasts for the Adriatic storm tide of November 2019. None of the models successfully predicted the first and highest sea level peak on 12 November 2019, but HIDRA2, NEMO and SCHISM all give a better forecast than HIDRA1 whose mean sea level does not even surpass the flooding threshold. As noted

in Cavaleri et al. (2020), this peak was difficult to forecast due to the delicate timing between the peak of winds and the peak of the full moon tide, combined with the formation of an unresolved local pressure disturbance over the west coast of Northern Adriatic. Relative timing of these influences turned out to be a *sine qua non* for a successful prediction – neither the winds nor pressure were, in themselves, in any way extraordinary. It is further shown in Cavaleri et al. (2020) that this particular storm tide could have been up to 25 cm higher had this scenario evolved 12 hours earlier when tidal peaks were themselves higher.

The peak on 13 November is slightly better predicted by maximum members of both HIDRAs than by NEMO or SCHISM, with HIDRA2 exhibiting a somewhat lower forecast spread than HIDRA1. Apart from this peak, all models captured the sea level variability quite well, which is in itself an implicit testament to the high skills of ECMWF atmospheric products.

Floods of December 2019 are another example of HIDRA2 superior performance over HIDRA1 and both ocean models in Koper. SSH observations and predictions in Koper during this period are depicted in Fig. 13. Several conclusions about

HIDRA2 behavior may be reached with regard to this particular flood. HIDRA2 ensemble appears to be closest to the obser-

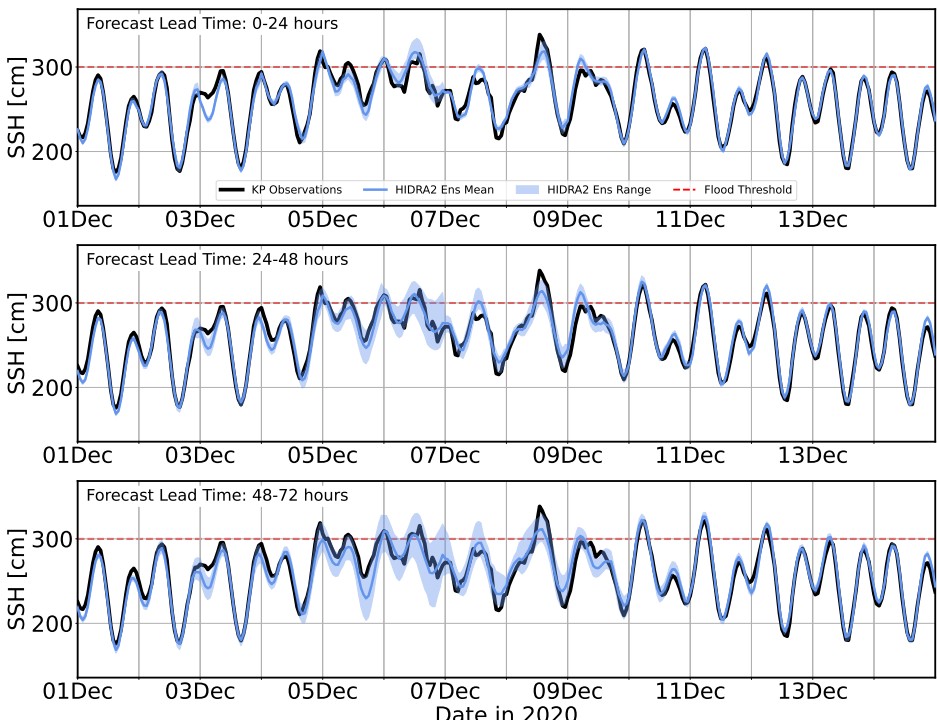

**Figure 14.** Comparison of HIDRA2 ensemble spread at forecast day 1 (top panel), day 2 (middle panel) and day 3 (bottom panel), during December 2020 in Koper. Semi-transparent regions in the plots depict the minimum-maximum range of the HIDRA2 ensemble.

vations and exhibits a substantially lower forecasting spread than the HIDRA1 ensemble. Low forecasting spread is acceptable when in conjunction with a well-behaved ensemble mean. In this case, the HIDRA2 ensemble mean is in excellent agreement with the observations. The same could be said for HIDRA1, albeit to a lesser degree. NEMO, however, completely misses the first two peaks between 15 and 17 December, slightly underestimates (like HIDRA1) the highest peak on 23 December, and

overall underestimates the minimum-maximum range of the sea level variations, corresponding to poorly predicted ebb levels after 23 December. SCHISM predicts the first two peaks but underestimates the peaks after 23 December. The vertical sea level range is much better captured by both HIDRAs, especially by HIDRA2. This result is consistent with our demonstration that HIDRA2 exhibits the lowest error in both the high and the low tail of sea level distributions (Fig. 11).

To inspect the behavior of the ensemble forecast spread, three timeseries were created from daily (72 h long) forecasts during

evaluation time window between 01 June 2019 and 31 December 2020. The first timeseries was constructed by concatenating each first day (i.e., 1–24 h of forecast) from each of the daily forecasts, thus containing predictions with lead times of 1–24 h on each respective day in the evaluation time window. The second and the third timeseries were constructed by concatenating 25–48 h (49–72 h) of forecast on each respective day in the evaluation time window. All three timeseries for the December 2020 floods are shown in Fig. 14. As expected, from the growing ensemble spread in the atmospheric forcing, HIDRA2 spread

is growing with forecast lead time as well. As we draw closer to a particular flooding event, the forecast spread drops, indicating an increased prediction certainty.

### 4.2.5 Spectral decomposition of forecasts during storms

To investigate the performance in geophysically relevant energy bands, we band-pass filtered the observed and the predicted SSH signals in energy bands, centered around four important periods: semi-diurnal tide (12 h period), diurnal tide (24 h period),
fundamental basin along-axis eigenmode (21.5 h period) and first excited along-axis eigenmode (10.9 h period).

Although incomplete, this SSH decomposition allows qualitative estimation of the excitation intensity of the basin eigenmodes during a particular storm, and also helps to qualitatively assign forecasting errors to specific frequency bands. However, since the amplitudes of filtered signals in Fig. 15 directly depend on the filter bandwidths, they should not be interpreted as direct contributions to the sea level due to respective geophysical phenomena (i.e., two tidal signals, two eigenmodes). They
should rather be read strictly as an additional insight into the model performance within a specific band with reference to filtered observations in the same band.

We applied a fifth-order Butterworth band-pass filter with the sampling rate of $(1\mathrm{h})^{-1}$. Low and high cutoff frequencies, which define the semi-diurnal filtering band $\Delta\omega_{12}$, were set to $\Delta\omega_{12} = [(12.5\,\mathrm{h})^{-1}, (11.5\,\mathrm{h})^{-1}]$. Similarly, diurnal cutoff frequencies were set to $\Delta\omega_{24} = [(24.5\,\mathrm{h})^{-1}, (23.5\,\mathrm{h})]^{-1}$. Fundamental seiche filtering band was estimated from Fig. 11 to be
$\Delta\omega_{21.5} = [(20\,\mathrm{h})^{-1}, (24\,\mathrm{h})^{-1}]$ which is also consistent with the seiche window used in Vilibić (2006). Finally, the first excited eigenmode band is defined as $\Delta\omega_{10.9} = [(11.4\,\mathrm{h})^{-1}, (10.5\,\mathrm{h})^{-1}]$. An example of this decomposition for November 2019 is shown in Fig. 15. For brevity we only show results for the NEMO model in the main body of the paper.

Identical analysis and related figures for the SCHISM model are available in the supplementary material to this paper. They illustrate that SCHISM exhibits very solid performance in the seiche energy bands.

All models exhibit an underestimation of the amplitude but are otherwise in phase with the observations in the $\Delta\omega_{12}$ band. In $\Delta\omega_{24}$, NEMO seems to be performing very well, with HIDRA2 slightly underestimating the range of the signal in this band. In the band $\Delta\omega_{21.5}$ NEMO is again closest to filtered observations while both HIDRA models overpredict the vertical range of the observed signal. Band $\Delta\omega_{10.9}$ is underpredicted in all models, but seems best (or rather least poorly) resolved by HIDRAs, with NEMO additionally exhibiting a substantial phase shift in the signal.

In any case, since both tidal bands and the ground state seiche are reliably predicted by all models, the reason for the forecasting errors must lie in the higher frequency bands with periods below 10.9 hours. This seems consistent with the occurrence of highly transient and localized low pressure over Venice mentioned in Cavaleri et al. (2020) and will be the subject of further research.

Similar remarks can be made regarding the December 2019 coastal flooding, depicted in Fig. 16. This event marked a sub-
optimal performance of NEMO, which is systematically underestimating SSH peaks and the overall vertical range of the SSH variability during this time window (Fig. 16, top panel). This caused NEMO to miss four floods out of eight. HIDRA models perform better, with HIDRA2 most reliably predicting all the flood peaks, most notably those on 15, 16 and 23 December 2019.

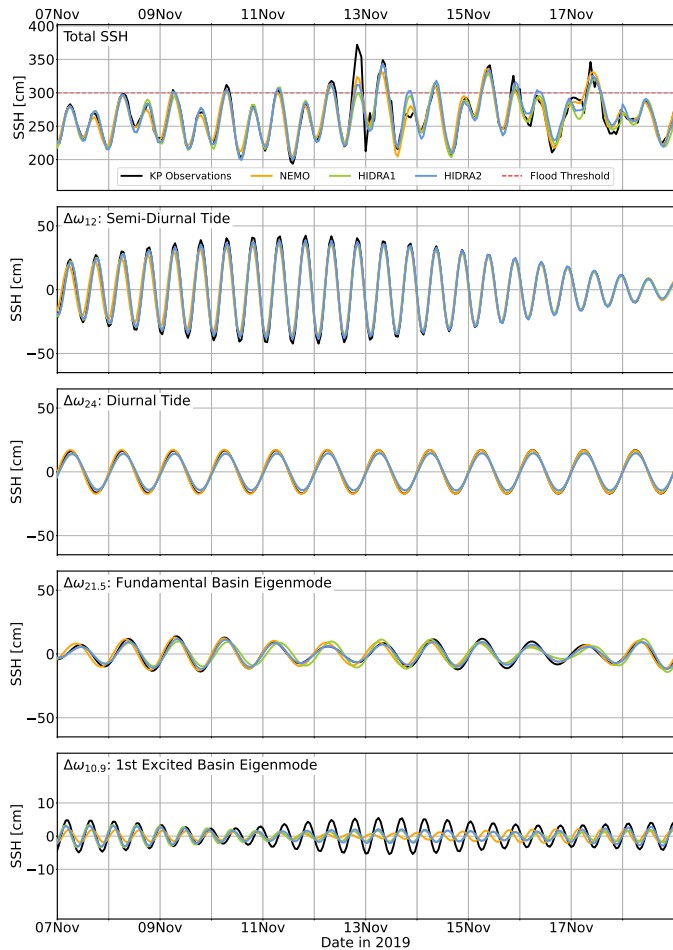

**Figure 15.** Comparison of total Koper SSH observations and forecasts (top panel) and their Band-Pass Filtered signals (bottom four panels) over four bands, centered around four geophysically relevant periods (semi-diurnal and diurnal tides and two lowest along-axis basin eigenmodes). Time window of the SSH signal spans from 7 November 2019 to 19 November 2019. Note the different vertical scale in the bottom $\Delta\omega_{10.9}$ panel.

The second and third panels in Fig. 16 demonstrate that all models are reliable in the diurnal tidal band $\Delta\omega_{24}$ but that HIDRA2 overestimates the signal in $\Delta\omega_{12}$. Since the overall performance of HIDRA2 is the best of all three models, it is unclear whether overshoots in $\Delta\omega_{12}$ could be interpreted as compensations for the underestimations in the nearby $\Delta\omega_{10.9}$ band. The bottom two panels in Fig. 16, however, indicate that part of the modeling errors stem from their underestimation of the basin seiches.

In the $\Delta\omega_{21.5}$ band, the HIDRA2 predictions most closely resemble the observations, followed by HIDRA1 and then NEMO (which is most severely underestimating this part of the signal). HIDRA2 is also the most reliable method in $\Delta\omega_{10.9}$ – but it nevertheless systematically underestimates the observations. HIDRA1 and NEMO performances are significantly worse,

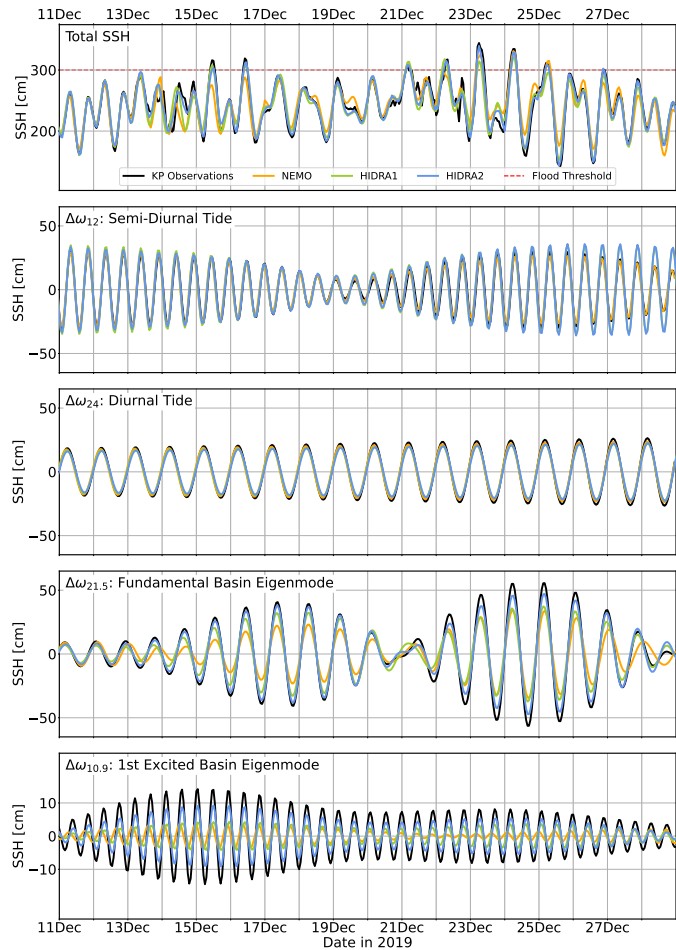

**Figure 16.** Same as Fig. 15, but for December 2019 coastal floods.

reaching one-half of the amplitude of HIDRA2 and one-third that of the observations. Poor performances of HIDRA1 and NEMO in $\Delta\omega_{21.5}$ and $\Delta\omega_{10.9}$ bands are simply another reflection of the fact depicted in Fig. 11, namely that both of these models struggle to generate an appropriate amount of energy in the bands around free oscillation eigenmodes.

## 445  5  Conclusions

This study presents a deep-learning based sea level model HIDRA2, suitable for operational sea level ensemble modeling due to its speed and accuracy. This work is a conceptual continuation of our previous attempt at sea level forecasting (Žust et al., 2021) and represents a substantial advancement over the first version (HIDRA1), setting a new state-of-the-art in machine learning SSH forecasting. The new architecture is validated by extensive ablation studies. The performance is benchmarked
against the current state-of-the-art Mediterranean forecasting setup of NEMO ocean model (available as part of Copernicus

Marine Service) and against a multi-decadal reanalysis run of the SCHISM model (Toomey et al., 2022) on an unstructured grid with very high coastal resolution. We demonstrate that HIDRA2 outperforms HIDRA1 as well as numerical ocean models across all sea level bins. We further show that HIDRA2 very accurately represents the energy contents in the bands around relevant geophysical periods (diurnal and semi-diurnal tides, and the lowest two free oscillation basin eigenmodes).

Performance is analyzed during several historic storms. Spectral decomposition of the total sea level signal into bands centered around tides and basin seiches is carried out to assign modeling errors to specific energy bands of the predicted sea levels. HIDRA2 consistently exhibits high skill in exciting the ground state Adriatic basin seiche at the appropriate time and with the appropriate phase and amplitude.

HIDRA2 is a good example of how the entanglement of deep-learning and geophysics may lead to reliable and numerically cheap models, which are able to mimic complex physical phenomena on the level of the best numerical physical models. Nevertheless, several extensions could be additionally explored. One possible extension is data ingestion from several tide gauges along the Adriatic coast and verification of whether the prediction accuracy at individual locations improves in such a multi-point prediction setup. Another extension is the inclusion of real-time in-situ measurements such as synoptic observations, satellite scatterometer and wind measurements. It would be interesting to migrate HIDRA2 to other Mediterranean locations or other semi-enclosed basins like the Baltic Sea, the Red Sea or the Chesapeake Bay to investigate its generalization properties. These will be objects of our future research.

*Code and data availability.* Implementation of HIDRA2 and the code to train and evaluate the model is available in the Git repository https://github.com/rusmarko/HIDRA2 (last access: 9 November 2022). We also include HIDRA2 weights pretrained on 2006–2018 and predictions for all 50 ensembles on June 2019–December 2020. The persistent version of HIDRA2 source code is available at https://doi.org/10.5281/zenodo.7307365 (Rus et al., 2022a). Training and evaluation of the model were performed on the datasets available at https://doi.org/10.5281/zenodo.7304086 (Rus et al., 2022c). Sea level datasets employed in this paper are available at https://doi.org/10.5281/zenodo.7277108 (Rus et al., 2022b).

*Author contributions.* MR was the main designer of HIDRA2 and reimplemented HIDRA1 in PyTorch. MK led the machine-learning part of the research and contributed to the design of HIDRA2. ML provided the geophysical background relevant for the HIDRA2 design and led the geophysical part of the research. AF and ML prepared the atmospheric and sea level training and evaluation datasets. MR, MK and ML analyzed the results and wrote the paper. All authors contributed to the final version of the manuscript.

*Competing interests.* The authors declare that they have no conflict of interest.

*Acknowledgements.* The authors would like to thank Tim Toomey and Alejandro Orfila for providing SCHISM sea level reanalysis time series for Koper location, which was used for benchmarking HIDRA2 in this work. We further thank the reviewers for taking the time to review the manuscript and for their constructive remarks which led to an improved paper. ML acknowledges the financial support from the Slovenian Research Agency (research core funding No. P1-0237).

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
