# Peer review of "HIDRA2: deep-learning ensemble sea level and storm tide forecasting in the presence of seiches – the case of Northern Adriatic"

_EGUsphere, 2022_

## Author Comment (AC1)

Dear editor, dear reviewer.

We thank the reviewer for their encouraging remarks. We have added further explanations to many sections of the manuscript and we would like to make a very short synopsis of what was done during this revision.

1. We have found a minor date-matching error in our preprocessing scripts that had to be corrected. This required a full re-training of all HIDRA2 versions, including ablations and HIDRA1. While the numbers in Tables 1 and 2 are slightly altered due to retraining, the relative performance of HIDRA2 is negligibly altered by this change and all conclusions regarding performance from the original manuscript still hold.
2. During the code review, we noticed that F1, precision, and recall scores are not correctly calculated due to an error at the boundaries of each prediction and we corrected the code. The correction does not change the conclusions regarding performance made in the original manuscript. We have updated and published the code for the calculation of the scores.
3. We have thoroughly rewritten and expanded the section of the ablation study.
4. We have obtained SSH timeseries from the high resolution SCHISM model and repeated all of the analyses with these timeseries. As the reviewer suspected, SCHISM performs better than the forecasting run of NEMO, but HIDRA2 still performs better than both at the tide gauge in Koper.
5. Terminology is now consistent with Gregory et al., as suggested.

---

## Author Comment (AC2)

Dear editor, dear reviewer.

Before we proceed with point-by-point addressing of the issues raised by the reviewer, we would like to make a very short synopsis of what was done during this revision.

1. We have found a minor date-matching error in our preprocessing scripts that had to be corrected. This required a full re-training of all HIDRA2 versions, including ablations and HIDRA1. While the numbers in Tables 1 and 2 are slightly altered due to retraining, the relative performance of HIDRA2 is negligibly altered by this change and all conclusions regarding performance from the original manuscript still hold.
2. During the code review, we noticed that F1, precision, and recall scores are not correctly calculated due to an error at the boundaries of each prediction and we corrected the code. The correction does not change the conclusions regarding performance made in the original manuscript. We have updated and published the code for the calculation of the scores.
3. We have thoroughly rewritten and expanded the section of the ablation study.
4. We have obtained SSH timeseries from the high resolution SCHISM model and repeated all of the analyses with these timeseries. As the reviewer suspected, SCHISM performs better than the forecasting run of NEMO, but HIDRA2 still performs better than both at the tide gauge in Koper.
5. Terminology is now consistent with Gregory et al., as suggested.

We proceed to a detailed point-by-point response below.

**This manuscript presents a new deep-learning storm surge forecasting (HIDRA2) for the Northern Adriatic which is an "updated" version of a previous deep learning architecture HIDRA1. The authors show that HIDRA2 outperforms its previous version HIDRA1 and it also performs better that the Copernicus Mediterranean Reanalysis. The paper is well written, and I think it must be published after some minor changes.**

The authors thank the reviewer for their constructive remarks. We have done our best to amend the manuscript according to their suggestions. We respond point by point below.

**Major comment:**

**The authors compare their new HIDRA2 with CMEMS Reanalysis, that is a 3D model in a regular horizontal grid. I recommend to the authors to compare the performance of HIDRA2 against numerical simulations specifically design to determine the sea surface elevation. As examples, the authors may use the global simulation by Muis et al. (2020) which has, in the Mediterranean, a 1.25 km of coastal resolution; or they can use the newest available simulation performed by Toomey et al. (2022) specifically designed for the Mediterranean Sea with a coastal resolution of 200 m. The latest hindcast includes the wave setup component that maybe relevant when computing the total elevation in the studied area.**

We would like to thank the reviewer for bringing these simulations to our attention. The authors of Toomey et al. (2022) have kindly shared these simulations with us and we have repeated all the analyses as the reviewer suggested.

These simulations are now analyzed in the same way as with NEMO. Related results are included partly in the main paper and partly in the Supplementary material of the paper.

However, we would like to emphasize that we are not benchmarking on a CMEMS reanalysis product, but rather on a forecasting one. The reason for this is that we want to evaluate our method's forecasting skills and it is thus most consistent to compare it to another forecast, not to a reanalysis.

**Minor comments:**

**- The authors use the term storm surge to refer to the combined effect of atmospheric pressure, winds, and astronomic tides. I strongly recommend the authors to follow the terminology detailed in Gregory et al. (2019). In that paper, they write: "N9 Storm surge: The elevation or depression of the sea surface with respect to the predicted tide during a storm." And they also write: "Sea-surface height (SSH) can be greatly elevated during a storm by a storm surge, and the consequent extreme sea level is sometimes called a storm tide". So, the combination of storm surge + astronomical tide should be referred to as storm tide, if we decide to follow the definitions in Gregory et al. (2019).**

We thank the reviewer for bringing these definitions to our attention. We now use the definitions from Gregory et al. in the revised manuscript. Correspondingly, HIDRA2 does not forecast storm surges, but rather a sea surface height or, during extreme storms, storm tide. The revised version reflects these distinctions. We have furthermore added the following clarification to the Introduction:

> In this paper we will adhere to the terminology, proposed in (Gregory et al., 2019): (i) the term *sea level* will denote total time-varying local water depth at the tide gauge in Koper, (ii) the term *sea surface height* is the height of sea level above (or below) the reference ellipsoid, (iii) the term *storm surge* denotes the elevation or depression of the sea surface with respect to the predicted tide during a storm, and (iv) the term *storm tide* is the sea surface height, elevated during a storm by a storm surge.

**- Line 17: this line needs, at least, two references where I have included (Ref. XXXX): "Global mean sea level rise, related to anthropogenic climate change (Ref. XXXX), is causing a worldwide increase in coastal flooding frequency and is leading to a myriad of negative consequences for coastal communities, civil safety and economies (Ref. XXXX)."**

The paragraph has been modified as the reviewer suggested:

> Global mean sea level rise, related to anthropogenic climate change (Arias et al., 2021), is causing a worldwide increase in coastal flooding frequency (Taherkhani et al., 2020) and is leading to a myriad of negative consequences for coastal communities, civil safety and economies (Ferrarin et al., 2020). Shallow semi-enclosed coastal regional basins like Northern Adriatic

**- Fig. 1: Please change the title of the figure to "Adriatic Sea topo-bathymetry".**

The title has been changed as the reviewer suggested.

**- Line 35-36: put the citation to Medvedev between parentheses.**

Done.

**- Line 48: "HIDRA1 ensemble (Žust et al., 2021) is a million times faster than the operational numerical ocean model ensemble based on NEMO engine". Although it maybe true, it is not fair to compare the computational time of a 3D model that computes water dynamics (currents, temperatures, salinities, ...) in several vertical layers with HIDRA1 or HIDRA2 that gives SSH information only and in a single point. The authors should compare the computational time with, for example Muis et al. (2020) or Toomey et al. (2022), and multiply their**

**computational time by the number of nodes that the other two studies are computing.**

We agree. We made it clearer what we aim at with this remark – we were aiming at operational time constraints.

Our own ensemble setup of NEMO3.6 (used in our previous GMD paper on HIDRA1 but not in the present paper) is numerically expensive and time-consuming, so the forecasters and civil rescue obtain their daily forecasts shortly before noon each day. NEMO morning bulletins are thus mostly issued based on a NEMO ensemble run from the previous evening. HIDRA architectures on the other hand allow instantaneous forecast production (for a single point, Koper) as soon as we get ensemble and tide gauge data. This is an immediate benefit for the forecasting service and for civil rescue response.

It is true that HIDRA computes predictions for a single point, while NEMO computes the sea state evolution of the entire basin, but this does not change the fact that HIDRA forecast of sea level is timely and NEMO's forecast generally is not. Warning triggers need only a single point sea level prediction, which HIDRA provides very fast and our in-house NEMO setup does not.

**- Fig. 10: it is difficult to appreciate the differences between the different datasets. I would recommend the authors to apply a filter to the spectrum. There may be a newer reference but I usually do it following the Chapter 5- Time-series Analysis Methods (https://www.sciencedirect.com/science/article/pii/B978044450756350006X) from Data Analysis Methods in Physical Oceanography by Emery and Thomson.**

Thank you for this suggestion. Power spectrums, presented in Figure 10, are now filtered using a Savitzky-Golay 3rd order filter and we hope the reviewer agrees that the plots are now more readable and interpretable. Furthermore, NEMO and SCHISM power spectrums are now depicted in the revised version side-by-side:

[Figure]

**Some thoughts:**

**1) Could this system be scaled up and applied to estimate SSH values at a Mediterranean scale?**

Thank you for this question, which touches upon our ongoing research. Preliminary analyses of HIDRA2 sea level forecasts in Venice and Ancona indicate a solid performance at these stations. We have, however, not yet tested locations outside the Adriatic basin. The answer therefore seems to be "yes" – HIDRA can probably scale to the Mediterranean basin. We cautiously expect it to perform well in other basins (the Baltic, the Northwest shelf, the German Bight, etc.) as well.

**2) How does the predictability change if the system is fed with the first predicted day?**

Thank you for this question. Following the reviewer suggestions, we have also benchmarked the case where the models are evaluated on the first day of predictions only. Details of the analysis are below, but unsurprisingly prediction errors for the first day are lower than when considering all three days. MAE, RMSE for HIDRA2 for the 3-day forecast are MAE = 4.12 cm and RMSE = 5.82 cm, while for the first day they are MAE = 3.38 cm and RMSE = 4.92 cm. Detailed statistics are available below. For brevity, we however chose not to include this in the manuscript itself.

**3-day forecast:**

**NEMO**

**Overall: MAE: 6.54, RMSE: 8.52**, bias: -1.23, acc: 79.14
**Storm tide events: MAE: 13.03, RMSE: 17.09**, bias: -11.24, acc: 49.68, recall: 63.58, precision: 100.00, F1: 77.73

SCHISM

**Overall: MAE: 5.57, RMSE: 7.50**, bias: 0.20, acc: 85.06
**Storm tide events: MAE: 11.04, RMSE: 14.70**, bias: -6.19, acc: 57.63, recall: 78.81, precision: 89.47, F1: 83.80

HIDRA1

**Overall: MAE: 4.72, RMSE: 6.73**, bias: -0.26, acc: 90.04
**Storm tide events: MAE: 12.95, RMSE: 17.65**, bias: -10.66, acc: 53.76, recall: 74.17, precision: 94.12, F1: 82.96

HIDRA2

**Overall: MAE: 4.12, RMSE: 5.82**, bias: 0.21, acc: 92.89
**Storm tide events: MAE: 9.77, RMSE: 14.07**, bias: -5.99, acc: 64.52, recall: 84.11, precision: 91.37, F1: 87.59

**1-day (first day) forecast:**

**NEMO**

**Overall: MAE: 6.37, RMSE: 8.33**, bias: -1.21, acc: 80.63
**Storm tide events: MAE: 12.11, RMSE: 16.36**, bias: -10.80, acc: 55.84, recall: 64.00, precision: 100.00, F1: 78.05

SCHISM

**Overall: MAE: 5.14, RMSE: 7.01**, bias: 0.18, acc: 87.45
**Storm tide events: MAE: 9.42, RMSE: 12.94**, bias: -4.57, acc: 63.64, recall: 82.00, precision: 91.11, F1: 86.32

**HIDRA1**

**Overall: MAE: 3.96, RMSE: 5.76**, bias: -0.23, acc: 93.55
**Storm tide events: MAE: 10.95, RMSE: 15.29**, bias: -8.61, acc: 59.74, recall: 82.00, precision: 100.00, F1: 90.11

**HIDRA2**

**Overall: MAE: 3.38, RMSE: 4.92**, bias: 0.09, acc: 95.53
**Storm tide events: MAE: 8.28, RMSE: 12.54**, bias: -4.83, acc: 70.78, recall: 88.00, precision: 93.62, F1: 90.72

**References:**

**Gregory, J.M., Griffies, S.M., Hughes, C.W. et al. Concepts and Terminology for Sea Level: Mean, Variability and Change, Both Local and Global. Surv Geophys 40, 1251–1289 (2019). https://doi.org/10.1007/s10712-019-09525-z**

**Muis S., Apecechea M. I., Dullaart J., de Lima Rego J., Madsen K. S., Su J., et al. (2020). A high-resolution global dataset of extreme sea levels, tides, and storm surges, including future projections. Front. Mar. Sci. 7. doi: 10.3389/fmars.2020.00263**

**Toomey, T., Amores, A., Marcos, M., & Orfila, A. Coastal sea levels and wind-waves in the Mediterranean Sea since 1950 from a high-resolution ocean reanalysis. Frontiers in Marine Science, 1873. doi: 10.3389/fmars.2022.991504**

---

## Author Comment (AC3)

Dear editor, dear reviewer.

Before we proceed with point-by-point addressing of the issues raised by the reviewer, we would like to make a very short synopsis of what was done during this revision.

1. We have found a minor date-matching error in our preprocessing scripts that had to be corrected. This required a full re-training of all HIDRA2 versions, including ablations and HIDRA1. While the numbers in Tables 1 and 2 are slightly altered due to retraining, the relative performance of HIDRA2 is negligibly altered by this change and all conclusions regarding performance from the original manuscript still hold.
2. During the code review, we noticed that F1, precision, and recall scores are not correctly calculated due to an error at the boundaries of each prediction and we corrected the code. The correction does not change the conclusions regarding performance made in the original manuscript. We have updated and published the code for the calculation of the scores.
3. We have thoroughly rewritten and expanded the section of the ablation study.

We proceed to a detailed point-by-point response below.

**This manuscript shows an implementation of a neural network (HYDRA2) to predict sea level at the Koper tidal station. This neural network is an improvement over HIDRA1 and it is compared to its predecessor and to a numerical ocean model NEMO. Particularly notable in this manuscript is the detailed validation of the performance of the model. I can recommend publishing this manuscript after minor changes.**

We thank the reviewer for encouraging comments.

**To facilitate the reading of the manuscript and the interpretation of the figures and table I would recommend the captions clarify if the authors show an independent validation (data not used during training and not used for the optimization of hyperparameters, if this is the case) or validation with dependent data. Likewise I think it would be useful to mention this also with the skill scores mentioned in the abstract (starting at line 7) whether these error reductions are obtained from the independent test data or not.**

We thank the reviewer for this remark. All validation is performed on an independent dataset that was hidden from the network during training. We now emphasize this fact throughout the revised manuscript to ensure clarity.

**I don't not have any doubts about the scientific soundness of the results, but adding this information would help readers understand the results of the manuscript more quickly.**

**Minor comments:**

**Line 5: "single member of ECMWF atmospheric ensemble": is this the central forecast or any single member (chosen at random)?**

We are using a fixed 42nd member of the atmospheric ensemble. Number 42 was chosen randomly to the extent that it is a tribute to the ultimate answer from the Hitchhiker's Guide to the Galaxy. Of course, over multi-year time intervals, this member is statistically completely equivalent to any other randomly selected member of the ECMWF ensemble prediction system. In other words, we could use any other ensemble member - or even choose a new random member in each run without substantially affecting the results. We now state this in the manuscript.

**Line 47: "HIDRA1 ensemble (Žust et al., 2021) is a million times faster than the operational numerical ocean model ensemble based on NEMO engine (Madec, 2016) at Slovenian Environment Agency": There is not a lot of context to understand this comparison. NEMO will provide you with a sea level estimate over the whole domain. Is this also the case for HIDRA1 or would it provide the sea-level for a single location?**

We thank the reviewer for this remark. Other reviewers raised the same issue and we have now amended the manuscript to better contextualize this comparison.

Our own ensemble setup of NEMO3.6 (used in our previous GMD paper on HIDRA1 but not in the present paper) is numerically expensive and time-consuming, so the forecasters and civil rescue obtain their daily forecasts shortly before noon each day. NEMO morning bulletins are thus mostly issued based on a NEMO ensemble run from the previous evening. HIDRA architectures on the other hand allow instantaneous forecast production (for a single point, Koper) as soon as we get ensemble and tide

gauge data. This is an immediate benefit for the forecasting service and for civil rescue response.

It is true that HIDRA computes predictions for a single point, while NEMO computes the sea state evolution of the entire basin, but this does not change the fact that HIDRA forecast of sea level is timely and NEMO's forecast generally is not. Warning triggers need only a single point sea level prediction, which HIDRA provides very fast and our in-house NEMO setup doesn't.

**Line 99: "HIDRA2 does not require explicit annotation of whether a location point belongs to land or sea, thus land masks are not generated." I am wondering if the land-sea mask would still be a useful feature to provide to the neural network as a wind over land would not generate seiches. I guess that the neural network compensates for this by learning the land-sea mask internally.**

We agree with the reviewer. Since the land mask is static, it is very likely that it is implicitly learned by the neural network, thus the network might not benefit from providing it at the input.

**Line 103: "three-days prediction lead time" I think that your ML model will give in one application the full 3-day time series. Can you confirm? Or do you rather need to apply the ML model iteratively to obtain the 3-day time series? Can you also clarify this in the manuscript?**

The reviewer is correct, HIDRA2 creates a 72-hour forecast in a single execution. We have made this clearer in the manuscript by adding the sentence that "A single prediction run of HIDRA2 model creates a 72-hour sea level timeseries for Koper location."

> SSH and regressed into the final SSH hourly predictions for the future 72 h by the Fusion-regression block (Sect. 3.1.3). A single prediction run of HIDRA2 model creates a 72-hour sea level timeseries for Koper location. Subsections below detail the individual blocks.

**104: "full ECMWF three-day forecast" -> "full" refers to the full ensemble (i.e. all ensemble members)?**

Yes, that is correct. We have now included "(i.e. all ensemble members)" behind "full".

**143: "prototype matching layer" Can you provide more information and a reference ?**

Prototype matching refers to a convolution between the convolutional kernel pattern and the encoded input data. Convolution is an application of a dot product between the two at each location. The dot product at any location will be large if the feature at that location is similar to the convolution kernel and low otherwise. In this context, the learned convolutional kernels can be considered prototype patterns and the convolution operation as prototype matching. We have updated the manuscript to make the term clearer:

> kernel, stride 2 and 64 output channels[1]. A ReLU activation and Dropout layers are applied, followed by a convolutional layer with 512 $4 \times 5$ kernels, which are by size equal to the input, meaning that convolution is essentially a dot product between each kernel and the input. The operation yields a higher value if kernel is similar to the input, so we refer to it as a *prototype matching layer*. It extracts features from different spatial positions, thus producing a 512-dimensional feature vector per group, i.e., 24 temporal vectors of size 512. The same processing architecture is applied to the pressure image sequence to produce 24

**section 4.1.1: this is an interesting and surprising result. Can the authors speculate why this is the case? (predicting full SSH leads to better results for extreme events). Could it be that the neural network internally limits its output range when working on anomalies? Do you expect this outcome to remain should one have more training data?**

We thank the reviewer for this question. Interpretability of neural networks is an open research problem, thus we can only speculate why HIDRA2 benefits from total sea levels but not so much from the residuals.

It seems that HIDRA2 learns to extract some information from the full sea level signal which consists of linear and nonlinear interactions between the tides and the storm surge. One of such nonlinear interactions, for example, reflects the fact that both storm surge and the tides modify local undisturbed water depths which impact their own barotropic propagation speeds and their respective topographic amplifications. Perhaps HIDRA2 is capable of resolving certain aspects of such interplays of phenomena. This interaction is practically non-existent during calm periods and is also less pronounced in the detided residual signal. This might explain why the benefit of full SSH is most obvious during storm tides. We have added the following discussion to the manuscript:

A possible explanation of this somewhat surprising behavior could perhaps be related to nonlinear interactions between tides and storm surges: both tides and storm surges modify local water depth which impacts their own barotropic wave propagation speeds and topographic amplifications, which ultimately define the onset time and the amplitude of any coastal flood in Koper. Such interactions are non-existent during calm conditions but they do play a role during stormy periods (Ferrarin et al., 2022). Perhaps HIDRA2 is able to anticipate certain aspects of nonlinear tide-surge couplings. This explanation is also consistent with the fact, detailed in Section 4.1.2, that among all atmospherically driven models the de-tided version HIDRA2$_{res}$ shows the worst performance during storm tide events, while versions incorporating tides come closest to HIDRA2 (see Fig. 8).

We are guessing that some further role is played by the fact that residual sea levels are contaminated with remnants of tidal signal but this contamination occurs in a very non-obvious way which is not related to the basin dynamics, but rather to our tidal algorithms, input data and (sea level - tide) subtraction.

**Section 4.1.2 "Ablation study": can you clarify that you retrained the network for the different test cases (without tides encoder, without atmospheric encoder,..) or you do rather zero-out the output of the corresponding encoder without re-training.**

Thank you for pointing this out: indeed we do leave out different input streams and/or encoders every time and we retrain a different network every time. Ablation study therefore consists of training and evaluating different architectures. It is not a post-processing activity.

We have now thoroughly rewritten, expanded, and restructured the Ablation section to make our actions more transparent.

**Typo:**

**Line 205: 1°/24 -> 1/24 °**

We changed this to (1/24)°.